# Mettl3 Regulates Osteogenic Differentiation and Alternative Splicing of Vegfa in Bone Marrow Mesenchymal Stem Cells

**DOI:** 10.3390/ijms20030551

**Published:** 2019-01-28

**Authors:** Cheng Tian, Yanlan Huang, Qimeng Li, Zhihui Feng, Qiong Xu

**Affiliations:** Guanghua School of Stomatology & Guangdong Provincial Key Laboratory of Stomatology, Sun Yat-sen University, 56 Ling Yuan Xi Road, Guangzhou 510055, China; tianch5@mail2.sysu.edu.cn (C.T.); huangylan@mail2.sysu.edu.cn (Y.H.); liqimeng22@outlook.com (Q.L.); fzhh136@sina.com (Z.F.)

**Keywords:** N6-methyladenosine, Mettl3, osteogenic differentiation, alternative splicing, Vegfa, PI3k-Akt

## Abstract

Bone mesenchymal stem cells (BMSCs) can be a useful cell resource for developing biological treatment strategies for bone repair and regeneration, and their therapeutic applications hinge on an understanding of their physiological characteristics. N^6^-methyl-adenosine (m^6^A) is the most prevalent internal chemical modification of mRNAs and has recently been reported to play important roles in cell lineage differentiation and development. However, little is known about the role of m^6^A modification in the cell differentiation of BMSCs. To address this issue, we investigated the expression of N^6^-adenosine methyltransferases (Mettl3 and Mettl14) and demethylases (Fto and Alkbh5) and found that Mettl3 was upregulated in BMSCs undergoing osteogenic induction. Furthermore, we knocked down *Mettl3* and demonstrated that *Mettl3* knockdown decreased the expression of bone formation-related genes, such as *Runx2* and *Osterix*. The alkaline phosphatase (ALP) activity and the formation of mineralized nodules also decreased after *Mettl3* knockdown. RNA sequencing analysis revealed that a vast number of genes affected by *Mettl3* knockdown were associated with osteogenic differentiation and bone mineralization. Kyoto encyclopedia of genes and genomes (KEGG) pathway analysis revealed that the phosphatidylinositol 3-kinase/AKT (PI3K-Akt) signaling pathway appeared to be one of the most enriched pathways, and Western blotting results showed that Akt phosphorylation was significantly reduced after *Mettl3* knockdown. Mettl3 has been reported to play an important role in regulating alternative splicing of mRNA in previous research. In this study, we found that *Mettl3* knockdown not only reduced the expression of *Vegfa* but also decreased the level of its splice variants, *vegfa-164* and *vegfa-188*, in *Mettl3*-deficient BMSCs. These findings might contribute to novel progress in understanding the role of epitranscriptomic regulation in the osteogenic differentiation of BMSCs and provide a promising perspective for new therapeutic strategies for bone regeneration.

## 1. Introduction

N^6^-methyl-adenosine (m^6^A) is a methylation modification at the N6 position of adenosine in both coding and noncoding RNAs and has been identified as the most prevalent internal chemical modification of mRNAs in eukaryotes [1,2]. Although its presence was first described in the 1970s, the function of m^6^A remained a mystery until recently. The biological function and significance of m^6^A modification have recently begun to enter the limelight as a result of an m^6^A RNA immunoprecipitation approach followed by high-throughput sequencing [3]. m^6^A modifications are frequently found near stop codons, at the beginning of 3′ untranslated regions (3′UTRs) and within internal long exons [4,5]. All m^6^A sites are found within sequences conforming to the consensus sequence RR m^6^A CH([G/A/U][G>A] m^6^A C[U>A>C]) [6]. The m^6^A modification occurs by a methyltransferase complex, including methyltransferase-like 3 (METTL3), methyltransferase-like 14 (METTL14) and Wilms’ tumor 1-associated protein (WTAP) in mammalian cells [7,8]. The modification is removed by fat-mass and obesity-associated protein (FTO) or α-ketoglutarate-dependent dioxygenase alkB homolog 5 (ALKBH5) [9,10]. The addition of m^6^A affects different aspects of mRNA metabolism, such as RNA stability, translation, alternative polyadenylation and pre-mRNA splicing [9,11,12]. Previous studies have demonstrated that m^6^A methylation has a profound effect on cell differentiation, embryonic development and stress responses [13,14,15].

Mesenchymal stem cells (MSCs) are a heterogeneous population of stem cells that can be harvested from many different sources and differentiate into mesoderm-type lineages, including osteoblasts, chondrocytes, and adipocytes [16,17,18,19]. MSCs have been widely used in stem cell transplantation, gene therapy, tissue engineering, and immunotherapy [20,21,22]. Bone mesenchymal stem cells (BMSCs) are commonly isolated and characterized from bone marrow aspirates and have been extensively investigated as the most promising cellular source for bone regeneration for both research and clinical purposes [23,24,25,26,27,28]. Numerous studies have demonstrated the beneficial effects of BMSCs on bone regeneration and repair, including stimulation of their differentiation into osteoblasts, stimulation of angiogenesis, immunomodulatory effects, antiapoptotic effects on osteogenic lineage cells, and recruitment of host MSCs/progenitor cells [29,30,31,32,33,34,35]. BMSCs have high affinity to differentiate into osteogenic lineages, and their differentiation into osteoblast cells is a pivotal step during bone regeneration and repair [36,37,38,39,40,41,42].

Bone reconstruction is a complex process of overlapping phases, including inflammation, repair, and remodeling, and involves many intracellular signaling pathways and growth factors [43,44,45,46]. Multiple signaling pathways have been reported to be involved in the osteogenic differentiation of BMSCs, such as bone morphogenetic protein (BMP)-Smad pathways, Wnt/β-catenin pathways, and PI3K-Akt pathways [47,48,49,50,51]. Osteogenesis and angiogenesis are highly coupled in the process of bone formation, and angiogenesis is a prerequisite step for bone tissue regeneration [52,53,54]. In tissue-engineered bone, BMSCs not only directly differentiate into endothelial cells to participate in new blood vessel formation in the bone defect area but also secrete angiogenic factors, such as vascular endothelial growth factor (VEGF) and angiopoietin, to promote local angiogenesis. As one of the most potent inducers of angiogenesis, VEGF is particularly intriguing due to its dual role in bone and endothelial development and plays an important role in bone formation [55,56]. Previous studies have shown that VEGF has been successfully used for enhanced bone regeneration due to its capacity to promote both osteogenic and angiogenic differentiation [57,58]. The angiogenic activity of BMSCs has been suggested to contribute to their regenerative capability.

While much evidence has demonstrated that the osteogenic differentiation of BMSCs is intimately associated with multiple genetic factors, such as signaling molecules and growth factors, the epigenetic regulatory mechanisms underlying cell differentiation have attracted increasing attention [59,60,61]. Recent evidence has illustrated that epigenetic modifications, such as histone acetylation and DNA methylation, are involved in the cell differentiation process of BMSCs during bone regeneration [62,63]. As another layer of epigenetic regulation (RNA epigenetics), m^6^A modification has recently been reported to play important roles in cell function and the differentiation of embryonic stem cells (ESCs) and various cancer cell types [64,65,66]. However, little is known about the role of m^6^A modification in the cell differentiation of BMSCs. Whether m^6^A affects the angiogenic factor Vegf during the osteogenic differentiation process of BMSCs also remains to be determined. Therefore, the aims of this study were to investigate the effect of m^6^A methylation on the osteogenic differentiation of BMSCs and to determine its role in regulating Vegf.

## 2. Results

### 2.1. Selection and Identification of BMSCs

To confirm the type of the separated BMSCs, third-passage cells were identified by flow cytometry to recognize the surface markers. The results showed expression of CD29+, CD44+, CD90+, CD34−, and CD45− (Figure 1A,B), indicating that the BMSCs were of high purity. Moreover, the BMSCs were able to be differentiated into osteoblast-like cells and adipose-likes cell when cultured in differentiating induction medium, as determined by Alizarin Red S staining and Oil Red O staining, respectively (Figure 1C–E). These data confirmed the status of the isolated and cultured BMSCs.

### 2.2. Expression of m^6^A Methyltransferase and Demethylases in BMSCs Undergoing Osteogenic Differentiation

During the osteogenic differentiation of BMSCs, cells were cultured in osteogenic induction medium at the indicated time points, and alkaline phosphatase (ALP) activity and mineralized nodule formation were estimated after induction. As shown in Figure 2C, cells cultured with osteogenic induction medium displayed significantly higher ALP activity on Day 7 than on Day 0. Mineralized matrix deposition was higher after 21 days of osteogenic induction than after 0, 7, and 14 days (Figure 2A,B). The expression levels of several osteogenic markers were also upregulated, as shown by quantitative real time polymerase chain reaction (qRT-PCR) and Western blotting, further confirming the osteogenic differentiation of BMSCs (Figure 2D,E).

To investigate the role of m^6^A modification in the osteogenic differentiation potential of BMSCs, the expression patterns of an m^6^A methyltransferase (Mettl3) and demethylases (Fto, Alkbh5) were measured during the osteogenic induction of BMSCs. Both the mRNA and protein levels of Mettl3 increased after 7 and 14 days of osteogenic induction (Figure 2F,G). Although *Fto* increased at the mRNA level after 7 and 14 days, its protein level showed no significant difference. Alkbh5 showed no significant difference at either the mRNA or protein level. Accordingly, the expression pattern of Mettl3 during osteogenic differentiation was consistent with those of the mineralization-related markers.

### 2.3. Effect of Mettl3 Knockdown on the Osteogenic Differentiation Potential of BMSCs

To explore the function of Mettl3 in the osteogenic differentiation process of BMSCs, specific shRNAs were applied to knock down its expression in BMSCs. Quantification of lentiviral gene transfer efficiency in BMSCs was measured via the proportion of fluorocytes. A transfer efficiency up to 90% was achieved at 48 h after transfection (Figure 3A). The Mettl3 protein level exhibited an approximate 60% decrease in the shRNA group (#sh1) compared with the negative control group, suggesting that Mettl3 was effectively silenced in BMSCs (Figure 3B). Mettl3-sh1 was thus chosen for further experiments.

To investigate the differentiation potential of BMSCs after *Mettl3* knockdown, the expression levels of several osteogenic markers were measured. The results showed that *Mettl3* knockdown reduced the mRNA levels of *Alp* and *Ocn* in BMSCs after osteogenic differentiation induction for 7 and 14 days (Figure 3C). The mRNA and protein expression of Runx2 and Osterix also decreased in *Mettl3*-shRNA cells (Figure 3D).

To verify the effect of *Mettl3* knockdown on the mineralization potential of BMSCs, ALP activity and the formation of mineralized nodules were then examined in *Mettl3*-shRNA cells. ALP activity decreased on Day 7, and the formation of mineralized nodules was reduced on Days 7, 14, and 21 after osteogenesis induction (Figure 3E,F).

### 2.4. Differentially Expressed Genes in Mettl3-Knockdown BMSCs

To further determine the effect of Mettl3 on the differentiation of BMSCs, *Mettl3*-shRNA and *Mettl*-shCtrl cells undergoing osteogenic induction for 14 days were subjected to RNA sequencing. The genes that were differentially expressed following *Mettl3* knockdown with a Log2 ratio >2 or <−2 were revealed in the heat map (Figure 4A). Among these genes, 556 genes were upregulated and 641 genes were downregulated (Figure 4B). Furthermore, gene oncology (GO) enrichment and KEGG pathway analyses were utilized to cluster the biological processes and pathways that were differentially inhibited between shMettl3 and shCtrl cells. In the biological process analysis of genes downregulated by *Mettl3* knockdown with a Log2 ratio <−2, the significant GO terms were related to ossification, cell adhesion, bone mineralization, and positive regulation of cytosolic calcium ion concentration (Figure 4C). From KEGG pathway analysis of the genes downregulated by *Mettl3* knockdown, the top ten enriched pathways were shown in Figure 4D. Among the signaling pathways that are closely related to osteogenic differentiation, the phosphatidylinositol 3-kinase (PI3K)-Akt signaling pathway appeared to be one of the most enriched pathways.

To confirm the inhibition of the PI3K-Akt pathway following *Mettl3* knockdown in BMSCs, Western blotting was performed to evaluate the phosphorylation level of Akt. Compared to the control group, the group with *Mettl3* knockdown exhibited a significant decrease in Akt phosphorylation (Figure 4E), indicating that PI3K-Akt signaling was suppressed by Mettl3 knockdown in BMSCs during the osteogenic differentiation process.

### 2.5. Effect of Mettl3 Knockdown on the Expression of Vegfa and Its Splice Variants

The angiogenic factor VEGF is closely related to the osteogenic differentiation potential of BMSCs. The *Vegfa* expression in *Mettl3*-knockdown BMSCs was detected, and the results showed that *Mettl3* knockdown decreased the expression level of *Vegfa* during the osteogenic differentiation of BMSCs (Figure 5B).

Alternative splicing is the most common event in transcript isoforms. [67,68] Mettl3 has been reported to play an important role in regulating alternative splicing of mRNA [69,70,71]. Previous studies reported that the alternative splicing of *Vegfa* affects the osteogenic differentiation of BMSCs [72,73]. *Vegfa* contains three homologous splice variants: *Vegfa-120*, *Vegfa-164*, and *Vegfa-188* [74]. To analyze whether *Vegfa* and its alternative splicing are modulated by Mettl3 in BMSCs undergoing osteogenic differentiation, we performed RT-PCR to detect three *Vegfa* isoforms. The results indicated that *Mettl3* knockdown decreased the mRNA expression of *Vegfa-164* and *Vegfa-188* but did not significantly alter *Vegfa-120* mRNA level (Figure 5A,C–E).

## 3. Discussion

m^6^A is a prevalent epitranscriptomic modification in mRNAs and is widely conversed across single-cell organisms, plants and vertebrates [75,76,77]. m^6^A methylation occurs during nascent pre-mRNA processing by a methyltransferase complex consisting of METTL3, METTL14, and WTAP and is removed by FTO and ALKBH5 [9]. This modification has been shown to regulate the pluripotency and differentiation of ESCs and somatic cell reprogramming. m^6^A methylation on some transcripts in ESCs, particularly those encoding developmental regulators, blocks HuR binding and maintains ESC self-renewal capability [78]. Depletion of m^6^A in the mRNA of *Mettl3*^−/−^ ESCs limits naïve marker priming and differentiation competence, which leads to a “hyper”-naïve pluripotency phenotype [79]. Overexpression of *Mettl3* in mouse embryonic fibroblasts (MEFs) increases m^6^A levels and significantly increases the reprogramming efficiency by enhancing the expression of key pluripotent factors [80].

BMSCs are the most suitable and widely used progenitor cell population for bone regeneration due to their capacity for multilineage differentiation and the potential to increase osteoinduction and osteogenesis [19,22]. BMSCs are known to stimulate bone tissue regeneration, but the mechanism by which the effect occurs has not yet been elucidated. To explore the role of m^6^A-dependent RNA methylation in the osteogenic differentiation of BMSCs, BMSCs were cultured using osteogenic induction medium to establish an osteogenesis model. The expression levels of main m^6^A methyltransferase and demethylases, including Mettl3, Fto and Alkbh5, were then evaluated. The results showed that Mettl3 mRNA and protein levels increased after osteogenic induction, but the protein expression of Fto and Alkbh5 levels did not significantly differ with or without osteogenic differentiation. These findings indicated that Mettl3-dependent RNA methylation might be involved in the osteogenic differentiation of BMSCs.

METTL3 (also known as MTA70) is identified as the main methyltransferase critical for m^6^A methylation [81,82]. Deletion or overexpression of *METTL3* changes the total m^6^A methylation level, which has a direct effect on cell survival, stem cell maintenance, and lineage determination [83,84,85]. Depletion of the *Mettl3* homolog in *Arabidopsis thaliana* gives rise to plants with altered growth patterns and reduced apical dominance [86]. The inhibition of the Drosophila *Mettl3* homolog (Dm ime4) has been proven to inhibit oogenesis [87]. *METTL3* inhibition leads to a concomitant decrease in the cellular m^6^A level and apoptosis of human HeLa cells [88]. To elucidate the role of Mettl3 in the osteogenic differentiation of BMSCs, *Mettl3* was silenced in BMSCs, and the effect of *Mettl3* knockdown on cell differentiation was investigated in the present study. The results showed that *Mettl3* knockdown reduced the accumulation of osteogenic differentiation markers such as Runx2 and Osterix. The ALP activity and the formation of mineralized nodules were also decreased after Mettl3 knockdown. Differentially expressed genes in *Mettl3* knockdown cells were subjected to RNA sequencing, and the results suggested that many genes were regulated by *Mettl3* knockdown. GO term analysis indicated that downregulated genes were associated with ossification, cell adhesion, and bone mineralization. KEGG pathway analysis revealed that the signaling pathways associated with the downregulated genes, such as PI3K-Akt signaling pathway, AMPK signaling pathway, and PPAR signaling pathway, were mostly involved in protein digestion and absorption, regulation of lipolysis in adipocytes, and osteogenic differentiation. Among the signaling pathways that are closely relevant to the osteogenic differentiation, the PI3K-Akt signaling pathway appeared to be one of the most enriched pathways. The PI3K-Akt signaling pathway is known to play a regulatory role in the survival, proliferation, migration, and differentiation of rabbit and human MSCs [89,90]. PI3K-Akt signaling also acts as a crucial regulator in bone tissue metabolism and the osteogenic differentiation of BMSCs [91]. Our Western blotting results confirmed a significant reduction in Akt phosphorylation after *Mettl3* knockdown, which was consistent with the RNA sequencing results. The above data suggested that Mettl3 increases the osteogenic differentiation of BMSCs.

Reconstructing local microcirculation and increasing blood vessel density are essential changes for effective bone regeneration [41,92,93,94,95]. VEGF is the most attractive and well-characterized factor for the therapeutic induction of new blood vessel growth. VEGF can be secreted by different MSC populations in a paracrine manner and influences endothelial cell behavior [57,58]. This growth factor is not only involved in angiogenesis but is also implicated in the maturation of osteoblasts, ossification, and bone turnover. Therefore, VEGF is widely used to induce angiogenesis and osteogenesis in bone tissue engineering constructs [96,97,98]. In rodents, *Vegfa* contains three homologous splice variants: *120*, *164*, and *188* amino acids [74]. *Vegfa-120*, the shortest variant, is less effective in promoting MSC growth. *Vegfa-164* plays a regulatory role in improving MSC proliferation and chondrogenic differentiation, and *Vegfa-188* is more effective in promoting the osteogenic differentiation of MSCs [99]. The present study indicated that *Mettl3* inhibition decreased the expression level of *Vegfa* during the osteogenic differentiation of BMSCs. The low expression of *Vegfa* might inhibit the osteogenic differentiation of BMSCs in turn. *Mettl3* localizes predominantly in nuclear speckles, which are sites of mRNA splicing and storage [100]. Several studies have implied that Mettl3 plays an important role in regulating alternative splicing of mRNA [69,70,71]. To clarify whether Mettl3 affects the alternative splicing pattern of *Vegfa*, the expression levels of *Vegfa* variants were determined using RT-PCR in Mettl3-depleted BMSCs. The results demonstrated that Mettl3 knockdown did not significantly alter *Vegfa-120* mRNA level but significantly decreased the mRNA levels of *Vegfa-164* and *Vegfa-188*, which are related to MSC proliferation and differentiation. These findings indicated that Mettl3 knockdown limited the expression of *Vegfa* and its bone formation-related splice variants in osteoblast-induced BMSCs.

In conclusion, the present study estimated the expression pattern of m^6^A methyltransferase and demethylases during the osteogenic differentiation of BMSCs and demonstrated that Mettl3 is highly expressed in osteogenically differentiated BMSCs. Investigating the role of Mettl3 in regulating cell differentiation showed that the loss of Mettl3 suppressed the osteogenic differentiation potential of BMSCs. *Mettl3* knockdown also decreased the expression of *Vegfa* and its splice variants in BMSCs. These findings might contribute to novel progress in the role of the epitranscriptome in osteogenic differentiation and provide a promising perspective for the development of innovative therapeutic strategies for bone regeneration.

## 4. Materials and Methods

### 4.1. Cell Culture and Osteogenic Differentiation

BMSCs were obtained from the femurs and tibias of 2–3-week-old Sprague-Dawley male rats (Animal Center of Sun Yat-sen University) as described previously by Liu et al. [101]. BMSCs were seeded in 25-cm^2^ culture flasks and were maintained by Dulbecco’s Modified Eagle Medium: Nutrient Mixture F-12 (DMEM/F12, Gibco, Grand Island, NY, USA) supplemented with 10% fetal bovine serum (FBS; Gibco, Grand Island, NY, USA) and 1% penicillin/streptomycin (Invitrogen, Carlsbad, NM, USA). The medium was renewed every 3 days. When reaching 90% confluence, BMSCs were harvested using trypsin/EDTA (Gibco, Grand Island, NY, USA) and reseeded into dishes (10 cm in diameter). The expression levels of different cell surface markers, including CD29+, CD44+, CD90+, CD34− and CD45−, were determined by flow cytometry. Cells from the third passage were used in further experiments.

For osteogenic differentiation, BMSCs were cultured using osteogenic induction medium, which contained DMEM/F12 supplemented with 100 nmol/L dexamethasone, 50 g/L ascorbic acid and 10 nmol/L β-glycerophosphate (Sigma-Aldrich, St. Louis, MO, USA) for 7 and 14 days. Cells without osteogenic induction medium were used as the control.

### 4.2. Mettl3 Knockdown Using shRNA Transfection

Short hairpin RNA (shRNA) against *Mettl3* shRNA: AGTCACAAACCAGATGAAATA and a nonspecific shRNA construct were designed and cloned into a hU6-MCS-Ubiquitin-EGFP-IRES-puromycin vector. The recombinant lentiviral vector was transfected into 293FT cells. The supernatant was harvested after 48 h and transduced into BMSCs. The cells were then incubated in solution with 1 μg/mL puromycin (Sigma-Aldrich, St. Louis, MO, USA) for 2 weeks. The stable clones were then maintained in 0.5 μg/mL puromycin and were observed and counted with a fluorescence microscope to ensure that the transfection rate was over 90%, and the knockdown effect was confirmed using a Western blotting assay. The cells were independently divided into 3 groups: control (untransduced group), shCtrl (negative control group), and shMettl3.

### 4.3. Western Blot Analysis

The BMSCs were harvested using RIPA lysis buffer (Cell Signaling Technology, Boston, MA, USA) containing protease inhibitor cocktail (Cwbiotech, Beijing, China). Forty micrograms of protein were electrophoresed in 10% sodium dodecyl sulfate polyacrylamide gel electrophoresis and then transferred onto polyvinylidene fluoride membranes (PVDF) (Millipore, Billerica, MA, USA) in transfer buffer with 10% methanol. The membrane was blocked in TBST containing 5% nonfat milk for 1 h at room temperature and then incubated with the following primary antibodies overnight at 4 °C: Mettl3 (1:1000; Proteintech, Chicago, IL, USA), Fto (1:1000; Abcam, Cambridge, UK), Alkbh5 (1:1000; Proteintech, Chicago, IL, USA), Runx2 (1:1000; Abcam, Cambridge, UK), Osterix (1:1000; Abcam, Cambridge, UK), Akt and P-Akt (1:1000; Cell Signaling Technology, Boston, MA, USA), Gapdh (1:1000; Abcam, Cambridge, UK), and Vinculin (1:1000; Cell Signaling Technology, Boston, MA, USA). The membrane was then incubated with secondary antibodies (Abcam, Cambridge, UK) for 1 h at room temperature after washing. The blot showing antibody binding was developed with an enhanced chemiluminescence system (Millipore, Billerica, MA, USA) and captured via an ImageQuant LAS 4000 mini system (GE Healthcare Life Sciences, Illinois, NJ, USA). Band densities were quantified using ImageJ v1.47 software (National Institutes of Health, Bethesda, MD, USA).

### 4.4. Alizarin Red S Staining

To detect mineralized nodule formation, BMSCs seeded in 6-well plates were induced in osteogenic medium for 0, 7, and 14 days. After culturing, the cells were rinsed with PBS and fixed in 4% paraformaldehyde solution for 20 min, stained with 1% Alizarin Red S (GL Biochem, Shanghai, China) solution at room temperature for 10 min, and washed 5 times. Mineralized nodules were then photographed under an inverted phase contrast microscope (Axiovert 40; Zeiss, Jena, Germany).

### 4.5. Alkaline Phosphatase Activity Assay

Cells seeded in 24-well plates were cultured in osteogenic differentiation medium for 7 days. Alkaline phosphatase activity was determined utilizing an ALP assay kit (Nanjing Jiancheng Bioengineering Institute, Nanjing, China). Cells were lysed with 1% Triton X-100 for 30 min at 4 °C. The optical density (OD) value of the solution was quantified by measuring the absorbance at 520 nm by a microplate reader (Tecan, Hombrechtikon, Switzerland). The protein content was examined by a bicinchoninic acid protein assay (Beyotime Biotechnology, Haimen, China), and ALPase levels were standardized to the total protein concentration.

### 4.6. Quantitative Real-Time PCR (qPCR) and Reverse Transcription PCR (RT-PCR) Monitoring of mRNA Levels

Total RNA from BMSCs was extracted using TRIzol reagent (Invitrogen, Carlsbad, NM, USA). RNA samples were reverse-transcribed for cDNA synthesis using Superscript III Reverse Transcriptase (Invitrogen, Carlsbad, NM, USA). Subsequently, cDNA was used in PCR. PCR was performed and monitored using the Light Cycler 480. Samples were analyzed using primers listed in Table 1.

To determine the expression of Vegfa splice variants, semiquantitative RT-PCR followed by agarose gel electrophoresis was performed. The pair of primers generated a different sized product for each of the splicing forms of VEGF mRNA. The predicted PCR products for the major forms, Vegf-120, Vegf-164, and Vegf-188 were 431, 563, and 635 bp, respectively [102]. One microgram of RNA was prepared and subjected to reverse transcription as described above. Reverse-transcribed cDNA was used for PCR using Taq DNA polymerase (Invitrogen, Carlsbad, NM, USA). PCR products were electrophoresed using 1.5% agarose gels, and images were captured using the FluorChem™ Q system (Alpha Innotech, San Jose, CA, USA).

### 4.7. RNA Sequencing

Approximately 20 ng of Poly (A) RNA were purified from total RNA and converted to cDNA. The cDNA samples were sequenced with a HiSeq 2000 system (Illumina Inc., San Diego, CA, USA). The expression of transcripts was quantified as reads per kilobase per million reads (RPKM). Genes were considered differentially expressed genes when the expression showed twofold changes using RPKM. Gene Ontology (GO) and Kyoto Encyclopedia of Genes and Genomes (KEGG) analyses were performed with the Database for Annotation, Visualization and Integrated Discovery (DAVID: http://david.abcc.ncifcrf.gov/) to identify the significantly enriched pathways.

### 4.8. Statistical Analyses

All experiments were performed at least in triplicate. The data are shown as the mean ± standard deviation and were analyzed using the SPSS 20.0 software program (SPSS, Chicago, IL, USA). One-way analysis of variance was applied to compare the experimental groups. Differences were considered statistically significant differences for *p* < 0.05.

## Figures and Tables

**Figure 1 ijms-20-00551-f001:**
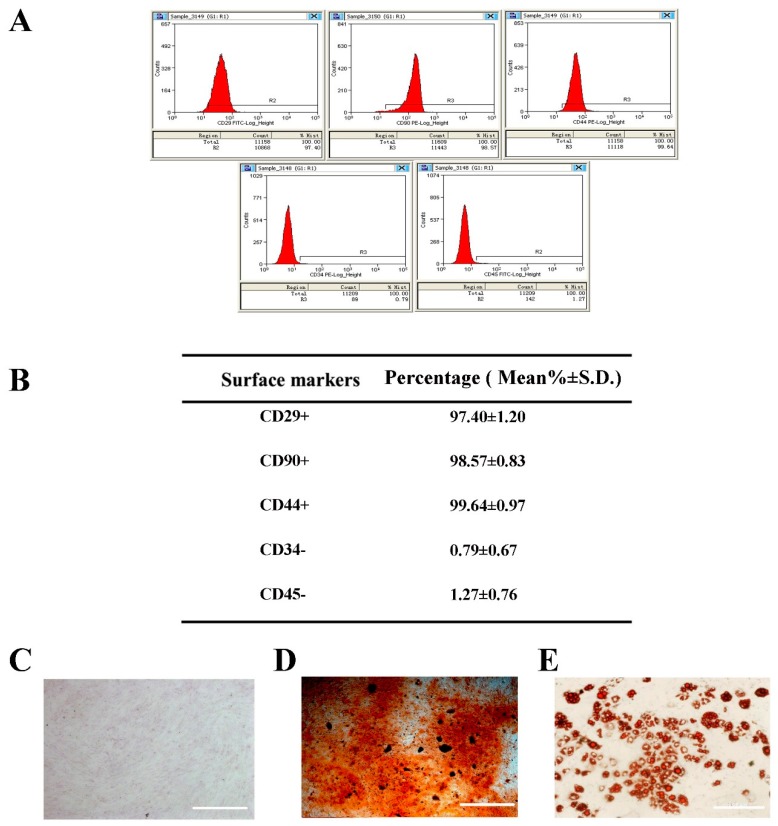
Characterization of BMSCs. (**A**) The expression of cell surface markers was evaluated by flow cytometry. Average of cells (%) are shown in (**B**) as Mean% ± S.D. (standard deviation). (**C**–**E**) The differentiation ability of BMSCs towards osteogenesis-like or adipogenesis-like cells was assessed by Alizarin Red S staining (**D**) or Oil Red O staining (**E**). The “white” scale bars represent 100 μm (original magnification ×100).

**Figure 2 ijms-20-00551-f002:**
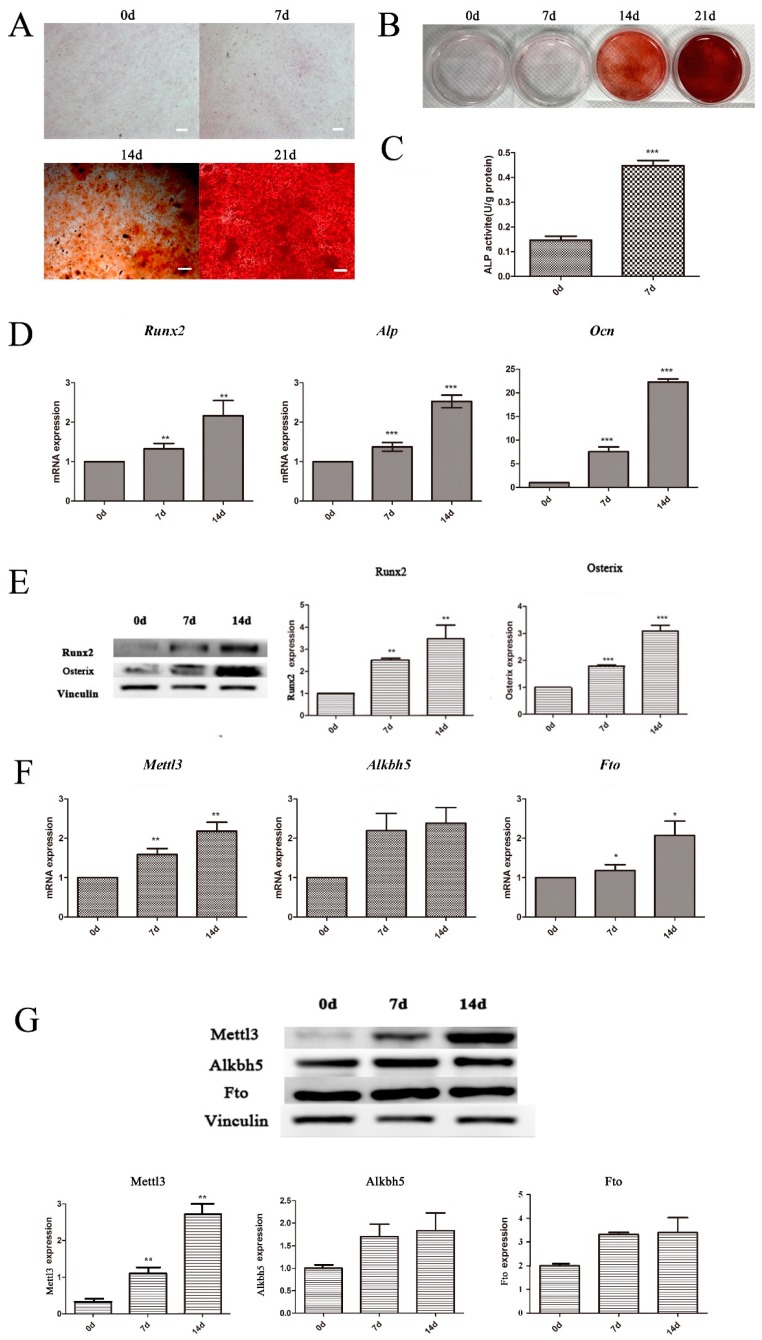
Osteogenic differentiation of BMSCs and expression of m^6^A methyltransferase and demethylases. (**A**,**B**) The formation of mineralized nodules was analyzed in BMSCs undergoing osteogenic differentiation on Days 7, 14, and 21. The “white” scale bars represent 100 μm (original magnification ×100). (**C**) ALP activity assays. ALP activity was significantly higher in the cells cultured in osteogenic differentiation medium for seven days. (**D**) The expression of *Runx2*, *Alp* and *Ocn* was assessed using qRT-PCR on Days 7 and 14 of culture in osteogenic induction medium. *Gapdh* was used as an internal control. (**E**) The expression of Runx2 and Osterix was examined using Western blotting. Vinculin was used as an internal control. The band intensities were analyzed using ImageJ software. (**F**,**G**) RNA methylation modification-related enzymes were detected using qRT-PCR and Western blotting in cells cultured in osteogenic induction medium for 7 and 14 days. *Gapdh* was used as an internal control for qRT-PCR. Vinculin was used as an internal control in Western blotting. All of the results represent the mean ± standard deviation of three independent experiments (*n* = 3). Significant difference compared with the control (* *p* < 0.05; ** *p* < 0.01; *** *p* < 0.001).

**Figure 3 ijms-20-00551-f003:**
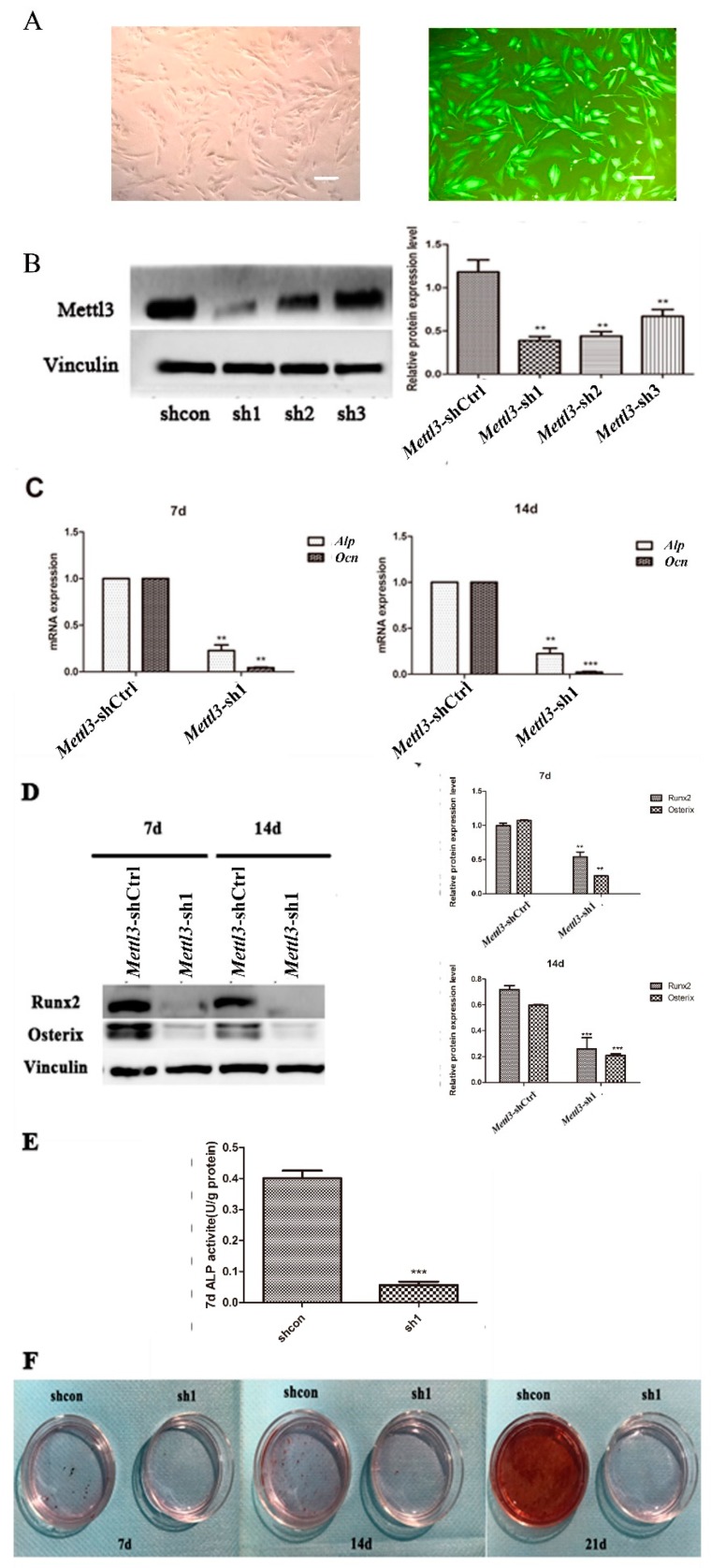
Effect of *Mettl3* knockdown on the osteogenic differentiation potential of BMSCs. (**A**) A green fluorescence protein marker was used to determine the transfer efficiency of *Mettl3* knockdown in BMSCs. After transfection for 72 h, the cells were observed under a microscope (on the left). The right image is an immunofluorescence image taken at the same time. The “black” scale bars represent 100 μm (original magnification ×100). (**B**) The expression level of Mettl3 was determined using Western blotting in the *Mettl3*-shRNA and *Mettl3*-shCtrl groups. Vinculin was used as an internal control. The band intensities were analyzed using ImageJ software. (**C**) The mRNA expression levels of *Alp* and *Ocn* in the *Mettl3*-shRNA and *Mettl3*-shCtrl groups were assessed using qRT-PCR after 7 and 14 days of osteogenic induction. *Gapdh* was used as an internal control. (**D**) The protein levels of Runx2 and Osterix in the *Mettl3*-shRNA and *Mettl3*-shCtrl groups were assessed using Western blotting after 7 and 14 days of osteogenic induction. Vinculin was used as an internal control. The band intensities were analyzed using ImageJ software. (**E**) ALP activity was determined in *Mettl3*-shRNA and *Mettl3*-shCtrl cells cultured in osteogenic differentiation medium for seven days. (**F**) The formation of mineralized nodules was analyzed in the *Mettl3*-shCtrl and *Mettl3*-shRNA groups undergoing osteogenic induction on Days 7, 14, and 21. Mineralization was analyzed using Alizarin Red S staining. All of the results represent the mean ± standard deviation of three independent experiments (*n* = 3). Significant difference compared with the control (* *p* < 0.05; ** *p* < 0.01; *** *p* < 0.001).

**Figure 4 ijms-20-00551-f004:**
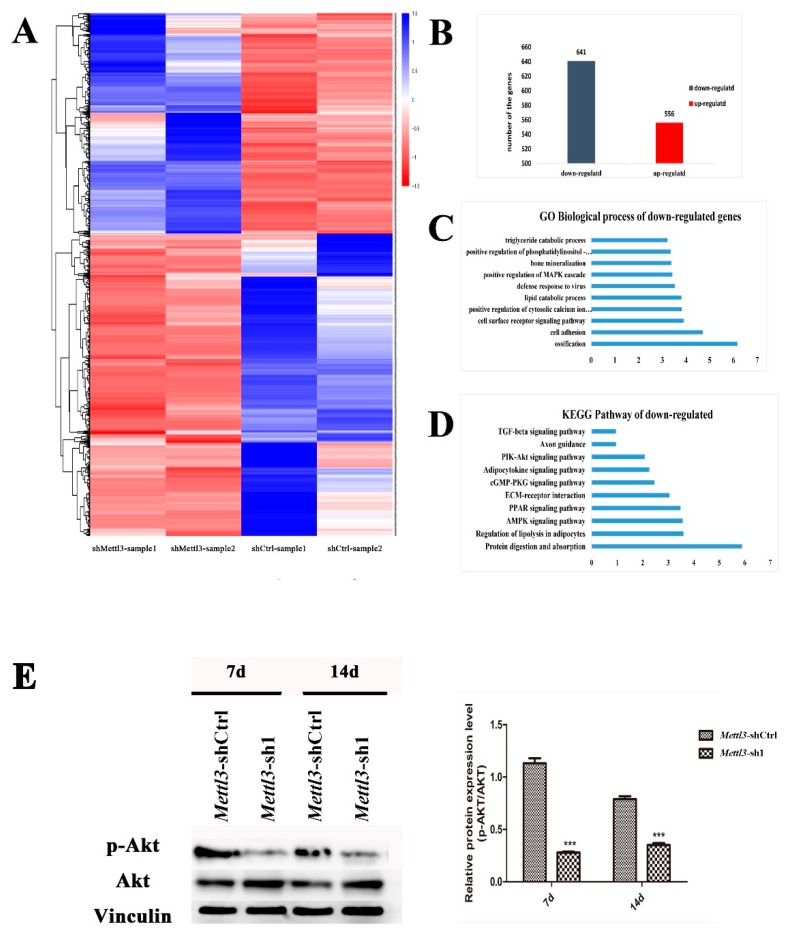
Differential expression of mRNAs between *Mettl3*-knockdown and control. (**A**) The differentially expressed genes in *Mettl3*-knockdown BMSCs under osteogenic induction medium are shown in the heat map. A *p* value cut-off of 0.05 and a Log2 ratio >2 or <−2 were used as a filter to identify the differentially expressed genes. (**B**) The differentially expressed genes were counted; among these genes, 556 were upregulated and 641 were downregulated. (**C**,**D**) GO and KEGG pathway analyses of genes that were differentially expressed with a Log2 ratio <−2, ordered according to gene enrichment. (**E**) The expression levels of total Akt and activated Akt (P-Akt) were assessed using Western blotting. Vinculin was used as an internal control. The band intensities were analyzed using ImageJ software. All of the results represent the mean ± standard deviation of three independent experiments (*n* = 3). Significant difference compared with the control (*** *p* < 0.001).

**Figure 5 ijms-20-00551-f005:**
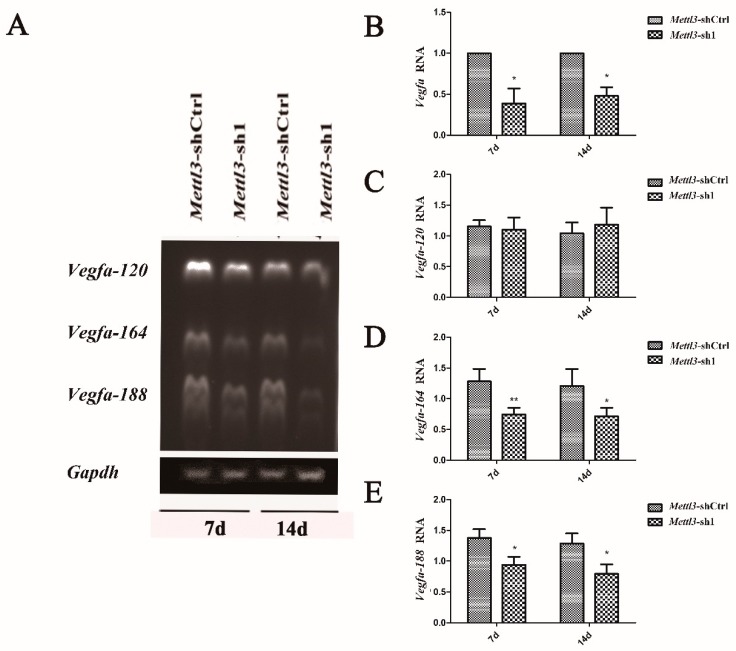
Effect of Mettl3 knockdown on the mRNA expression of *Vegfa* and its splice variants. (**A**) RT-PCR was performed to amplify mRNA products specific to *Vegfa* isoforms or an mRNA product specific to *Gapdh*. (**B**) The expression of *Vegfa* was decreased after Mettl3 knockdown, as indicated by the qRT-PCR results. (**C**–**E**) The mRNA expression of *Vegfa* isoforms was assessed with the expected mRNA sizes of 431 bp for *Vegfa-120*, 563 bp for *Vegfa-164*, and 635 bp for *Vegfa-188*. *Gapdh* was used as an internal control for qRT-PCR and RT-PCR. The data are presented as the mean ± standard deviation of three independent experiments (*n* = 3). Statistically significant difference relative to the shCtrl group (* *p* < 0.05; ** *p* < 0.01).

**Table 1 ijms-20-00551-t001:** Primers Used for the Analysis of mRNA Levels by qRT-PCR and RT-PCR.

Gene	Forward Primer	Reverse Primer
*Ocn*	5′ ATCCATGCAGGCATCTCACC 3′	5′ ACCTAACCAATTGCCCCCAG 3′
*Alp*	5′ TCGATGGCTTTGGTACGGAG 3′	5′ TGCGGGACATAAGCGAGTTT 3′
*Runx2*	5′ GGCCAGGTTCAACGATCTGA 3′	5′ GGACCGTCCACTGTCACTTTA 3′
*Mettl3*	5′ CTTTAGCATCTGGTCTGGGCT 3′	5′ CCTTCTTGCTCTGCTGTTCCT 3′
*Alkbh5*	5′ ACCACCAAACGGAAGTACCAG 3′	5′ TCATCCTGGCTGAAGAGACG 3′
*Fto*	5′ ACTGGTTTTCCGAGAGGCTG 3′	5′ GTGAGCACGTCTTTGCCTTG 3′
*Vegfa*	5′ CTGCTCTCTTGGGTGCACTGG 3′	5′ CACCGCCTTGGCTTGTCACAT 3′

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
