# Peer review of "Mettl3 Regulates Osteogenic Differentiation and Alternative Splicing of Vegfa in Bone Marrow Mesenchymal Stem Cells"

_ijms, 2019, doi:10.3390/ijms20030551_

Round 1
Reviewer 1 Report
The manuscript by Cheng et al., aims to address the effect of one of most prevalent internal chemical modification of mRNA, the N6-methyl-adenosine (m6A), on osteogenic differentiation of Bone Mesenchymal Stem Cells (BMSCs) and on angiogenesis regulating alternative splicing of Vegf.
The authors reveal the involvement of m6A methyltransferase on osteogenic differentiation evaluating some parameters: ALPase activity, mineralized matrix deposition and the induction of some osteogenic markers. All the results showed in the manuscript are congruent with the involvement of METTL3 in osteogenic differentiation although the expression of demethylases genes and proteins (Fto and Allbh5) display a contradictory trend. In Figure 2F and 2G the authors have seen both the induction of Mettl3 and of Alkbh5 and Fto during differentiation even if they underline that the increase of Fto and Alkbh5 proteins, after 7 and 14 days of induction of osteogenic differentiation, do not it's significant. These data are not convincing above all observing the significative induction of Fto mRNA level after 14 days of BMSCs differentiation but also observing the error bars relative to band intensities of western blot. How many samples show the same trend? Could the authors give strength to their results?
In Figure 3 the authors investigate successfully the differentiation potential of BMSCs after shMettl3 lentiviral transfection. To better appreciate the western blot showed in figure 3B, the authors should choose a less exposed band for the internal control: Vinculin, since the bands seem saturated to be correctly analyzed using imageJ. Relative to experiments in figure 3 it would be interesting investigate the effect of Mettl3 knockdown during the osteogenic differentiation of BMSCs also on mRNA and protein expression levels of demethylases (Fto and Alkbh5).
The authors can check the expression levels of demethylase mRNAs in the RNA sequencing data showed in figure 4.
In the last paragraph are lacking some references on previous studies regarding the alternative splicing regulation by Mettl3 and how the Vegfa different spliced forms are able to affects in a different way the osteogeinc differentiation of BMSCs.
The overall study seems very interesting for the development of novel therapeutic strategies for bone regeneration moreover studying the epi-transcriptome should useful for other cells and tissues and its regulation and manipulation can be suitable for regenerative medicine in general.
Author Response
Authors’ Point-by-Point Responses to the Editor and Reviewers
Dear Editor,
We would like to thank you and the reviewers for providing us with very valuable and helpful comments on our manuscript. All the authors have had a chance to revise and improve this manuscript, and our final revision has been drafted according to the comments from the editor and reviewers. All revisions are highlighted in red in the manuscript.
Reviewer comments:
Reviewer: 1
Comments to the Author
The manuscript by Cheng et al., aims to address the effect of one of most prevalent internal chemical modification of mRNA, the N6-methyl-adenosine (m6A), on osteogenic differentiation of Bone Mesenchymal Stem Cells (BMSCs) and on angiogenesis regulating alternative splicing of Vegf.
Comments:
1. The authors reveal the involvement of m6A methyltransferase on osteogenic differentiation evaluating some parameters: ALPase activity, mineralized matrix deposition and the induction of some osteogenic markers. All the results showed in the manuscript are congruent with the involvement of METTL3 in osteogenic differentiation although the expression of demethylases genes and proteins (Fto and Allbh5) display a contradictory trend. In Figure 2F and 2G the authors have seen both the induction of Mettl3 and of Alkbh5 and Fto during differentiation even if they underline that the increase of Fto and Alkbh5 proteins, after 7 and 14 days of induction of osteogenic differentiation, do not it's significant. These data are not convincing above all observing the significative induction of Fto mRNA level after 14 days of BMSCs differentiation but also observing the error bars relative to band intensities of western blot. How many samples show the same trend? Could the authors give strength to their results?
We thank the reviewer for the comment.
The experiments were performed in triplicate, and averaged data were generated. For each parameter, we tested the results obtained with the three samples, i.e., replicated three times on the different cells, and obtained averaged data for the three different cell types. As shown in figure 2F and 2G, although Fto increased at the mRNA level after 7 and 14 days of osteogenic induction, its protein level showed no significant difference. Alkbh5 showed no significant difference at either the mRNA or protein level. The data of western blotting have been checked.The Fto data are shown as follows.
2. In Figure 3 the authors investigate successfully the differentiation potential of BMSCs after shMettl3 lentiviral transfection. To better appreciate the western blot showed in figure 3B, the authors should choose a less exposed band for the internal control: Vinculin, since the bands seem saturated to be correctly analyzed using imageJ. Relative to experiments in figure 3 it would be interesting investigate the effect of Mettl3 knockdown during the osteogenic differentiation of BMSCs also on mRNA and protein expression levels of demethylases (Fto and Alkbh5).
Thank you for your valuable advice.
Actually, the band of Vinculin seemed to be overexposed due to poor image processing. We have chosen a less exposed band for the internal control in figure 3B in the manuscript.
To determine the effect of Mettl3 knockdown on the differentiation of BMSCs, RNA sequencing was performed to analyze differentially expressed genes in Mettl3 knockdown cells in the present study. The data showed that Fto and Alkbh5 were not differentially expressed after Mettl3 knockdown. The result was further confirmed by qRT-PCR (Figure 1).
Figure 1 Effect of Mettl3 knockdown on the expression of Fto and Alkbh5. The mRNA expression levels of Fto and Alkbh5 in the Mettl3-shRNA and Mettl3-shCtrl groups were assessed using qRT-PCR. Gapdh was used as an internal control.
3. The authors can check the expression levels of demethylase mRNAs in the RNA sequencing data showed in figure 4.
We thank the reviewer for the suggestion.
We checked the expression levels of demethylases (Fto and Alkbh5) mRNAs in the RNA sequencing data and found Fto and Alkbh5 were not differentially expressed after Mettl3 knockdown, as shown below.
4. In the last paragraph are lacking some references on previous studies regarding the alternative splicing regulation by Mettl3 and how the Vegfa different spliced forms are able to affects in a different way the osteogeinc differentiation of BMSCs.
Thank you for your valuable advice. We have added relevant references to the manuscript according to the reviewer’s suggestion. We also described how the Vegfa different spliced forms affect the osteogeinc differentiation of BMSCs in different ways in the fourth paragraph of the discussion. The revisions are highlighted in red in the manuscript.

Reviewer 2 Report
Review of Manuscript with title: Mettl3 regulates osteogenic differentiation and alternative splicing of Vegfa in bone marrow mesenchymal stem cells.
The above titled manuscript by Cheng and colleagues investigated the effect of m6A methylation on the osteogenic differentiation of BMSCs and to determine its role in regulating VEGF. This manuscript is very timely and deserves to be published. The human epigenome plays a key role in determining cellular identity and eventually function. All somatic cells contain the same genetic material but display varied cellular phenotypes. Phenotypic variability is determined not only by genome but also the epigenome. The interaction of the genome and the epigenome is universal in all multicellular organisms and necessary for developmental biology, differentiation and function. Thus this study looking at the role epigenomic modifications play in MSCs differentiation is spot on. It is true that both the genome and epigenetic modifications are likely to influence MSCs differentiation. Thus this study is very important.
Overall l recommend publication of the manuscript after effecting ALL minor changes shown below in FULL.
Comments and Revisions needed
1. Font must be the same throughout the manuscript
2. Authors must differentiate the names of the genes when talking about human and rodent genes. For example naming of VEGF must be specific. One format must be followed. For example for human VEGF it must be italicised and capitalised. Authors must check gene names throughout the manuscript.
3. M6A is not m6A. Please check throughout the manuscript
4. Authors used very few and old references when describing the characterisation of MSCs. For example reference 12 is from 1992. Reference 12 only talks about osteogenic differentiation of MSCs yet the authors used it for osteogenic, chondrogenic and adipogenic differentiation. New data has since come out to clearly show features of different MSCs such that these old references should NOT be used. In addition new papers have recently been published on regenerative medicine and tissue engineering. Authors used references from 2006 for regenerative medicine etc. Within the last 10 years a lot of new information has been published on regenerative medicine such that 2006 is like ages ago. Several sentences in the manuscript require more than one reference and these should be added. In addition, the manuscript is on epigenetics, MSCs and differentiation and yet has less than 100 references. These topics are hot topics nowadays and the necessary references must be added. The following references should be used in the manuscript with addition of necessary text before publication:
Li Q, Gao Z, Chen Y and Guan MX. (2017). The role of mitochondria in osteogenic, adipogenic and chondrogenic differentiation of mesenchymal stem cells. Protein Cell 8, 439-445.
Dzobo K, Thomford NE, Senthebane DA, et al. (2018). Advances in Regenerative Medicine and Tissue Engineering: Innovation and Transformation of Medicine. Stem Cells Int 2018, 2495848.
Dzobo K, Vogelsang M, Thomford NE, et al. (2016b). Wharton's Jelly-Derived Mesenchymal Stromal Cells and Fibroblast-Derived Extracellular Matrix Synergistically Activate Apoptosis in a p21-Dependent Mechanism in WHCO1 and MDA MB 231 Cancer Cells In Vitro. Stem Cells Int 2016, 4842134.
Van Zoelen EJ, Duarte I, Hendriks JM and Van Der Woning SP. (2016). TGFbeta-induced switch from adipogenic to osteogenic differentiation of human mesenchymal stem cells: identification of drug targets for prevention of fat cell differentiation. Stem Cell Res Ther 7, 123.
Xu Y, Li Z, Li X, et al. (2015). Regulating myogenic differentiation of mesenchymal stem cells using thermosensitive hydrogels. Acta Biomater 26, 23-33
5. Figure 1B must show data as average percentages (mean ± S.D).
6. The authors must explain why they never checked for pluripotency genes such as Oct4, Sox2 etc
7. Although the manuscript is on MSCs from male rats, data presentation, especially on MSCs characterisation, is very similar to: Dzobo K, Turnley T, Wishart A, et al. (2016a). Fibroblast-Derived Extracellular Matrix Induces Chondrogenic Differentiation in Human Adipose-Derived Mesenchymal Stromal/Stem Cells in Vitro. Int J Mol Sci 17. Authors would benefit from using such published manuscript for guidance when presenting their results. And should reference such papers.
8. Figure 3B, D. Mettl-3-sh3 did not affect Mettl3 protein levels very much yet affected osteogenic differentiation in the same way as Mettl3-sh1. Please authors must comment on this in the manuscript.
Once ALL these technical and minor issues are resolved this manuscript must be published. Such studies are needed to explain cellular processes that are not wholly due to genetic causes.
Thank you
Author Response
Authors’ Point-by-Point Responses to the Editor and Reviewers
Dear Editor,
We would like to thank you and the reviewers for providing us with very valuable and helpful comments on our manuscript. All the authors have had a chance to revise and improve this manuscript, and our final revision has been drafted according to the comments from the editor and reviewers. All revisions are highlighted in red in the manuscript.
Reviewer comments:
Reviewer: 2
Comments to the Author
1. Font must be the same throughout the manuscript
We thank the reviewer for the suggestion. We have carefully corrected the font throughout the manuscript.
2. Authors must differentiate the names of the genes when talking about human and rodent genes. For example naming of VEGF must be specific. One format must be followed. For example for human VEGF it must be italicised and capitalised. Authors must check gene names throughout the manuscript.
We are sorry for our incorrect writing on the names of the genes. We have carefully corrected the phrases throughout the manuscript. The revisions are highlighted in red in the manuscript.
3. M6A is not m6A. Please check throughout the manuscript
We have carefully corrected this phrase throughout the manuscript. All m6A have been changed to m6A.
4. Authors used very few and old references when describing the characterisation of MSCs. For example reference 12 is from 1992. Reference 12 only talks about osteogenic differentiation of MSCs yet the authors used it for osteogenic, chondrogenic and adipogenic differentiation. New data has since come out to clearly show features of different MSCs such that these old references should NOT be used. In addition new papers have recently been published on regenerative medicine and tissue engineering. Authors used references from 2006 for regenerative medicine etc. Within the last 10 years a lot of new information has been published on regenerative medicine such that 2006 is like ages ago. Several sentences in the manuscript require more than one reference and these should be added. In addition, the manuscript is on epigenetics, MSCs and differentiation and yet has less than 100 references. These topics are hot topics nowadays and the necessary references must be added. The following references should be used in the manuscript with addition of necessary text before publication:
Li Q, Gao Z, Chen Y and Guan MX. (2017). The role of mitochondria in osteogenic, adipogenic and chondrogenic differentiation of mesenchymal stem cells. Protein Cell 8, 439-445.
Dzobo K, Thomford NE, Senthebane DA, et al. (2018). Advances in Regenerative Medicine and Tissue Engineering: Innovation and Transformation of Medicine. Stem Cells Int 2018, 2495848.
Dzobo K, Vogelsang M, Thomford NE, et al. (2016b). Wharton's Jelly-Derived Mesenchymal Stromal Cells and Fibroblast-Derived Extracellular Matrix Synergistically Activate Apoptosis in a p21-Dependent Mechanism in WHCO1 and MDA MB 231 Cancer Cells In Vitro. Stem Cells Int 2016, 4842134.
Van Zoelen EJ, Duarte I, Hendriks JM and Van Der Woning SP. (2016). TGFbeta-induced switch from adipogenic to osteogenic differentiation of human mesenchymal stem cells: identification of drug targets for prevention of fat cell differentiation. Stem Cell Res Ther 7, 123.
Xu Y, Li Z, Li X, et al. (2015). Regulating myogenic differentiation of mesenchymal stem cells using thermosensitive hydrogels. Acta Biomater 26, 23-33
We thank the reviewer for the valuable suggestion.
We have added some relevant references to the manuscript with addition of necessary text according to the reviewer’s suggestion. Some old references have been replaced with new references. The revisions are highlighted in red in the manuscript.
5. Figure 1B must show data as average percentages (mean ± S.D).
We thank the reviewer for the suggestion. The data in Fig. 1B were shown as average percentages (mean ± S.D) according to the reviewer’s suggestion.
6. The authors must explain why they never checked for pluripotency genes such as Oct4, Sox2 etc.
We thank the reviewer for the valuable suggestion.
BMSCs comprise multipotent stem cells with the capacity to differentiate into mesoderm-type lineages, including osteoblasts, chondrocytes, and adipocytes [1-3]. The transcription factors Oct4 and SOX2 are essential for the maintenance of pluripotency in early embryos and embryonic stem cells (ESCs). The stemness-related genes play important roles in maintaining the proliferation and differentiation potential of BMSCs. [4-6]. The purpose of this study was to explore the effect of Mettl3-dependent RNA methylation on the osteogenic differentiation potential of BMSCs. Thus, the expression levels of several osteogenic markers and the mineralization potential of BMSCs were evaluated after Mettl3 knockdown. It would be interesting to investigate the effect of m6A modification on the expression of the pluripotent genes in BMSCs, which need further study.
References:
1. Li Q., Gao Z., Chen Y., Guan MX. The role of mitochondria in osteogenic, adipogenic and chondrogenic differentiation of mesenchymal stem cells. Protein Cell. 2017; 8:439-445.
2. Tu C., Xiao Y., Ma Y., Wu H., Song M. The legacy effects of electromagnetic fields on bone marrow mesenchymal stem cell self-renewal and multiple differentiation potential. Stem Cell Res Ther. 2018; 9:215.
3. Waldner M., Zhang W., James IB., Allbright K., Havis E., Bliley JM., Almadori A., Schweizer R., Plock JA., Washington KM., Gorantla VS., Solari MG., Marra KG., Rubin JP. Characteristics and immunomodulating functions of adipose-derived and bone marrow-derived mesenchymal stem cells across defined human leukocyte antigen barriers. Front Immunol. 2018; 9:1642.
4. Lee JH, Lee WJ, Jeon RH, Lee YM, Jang SJ, Lee SL, Jeon BG, Ock SA, King WA, Rho GJ. Development and gene expression of porcine cloned embryos derived from bone marrow stem cells with overexpressing Oct4 and Sox2. Cell Reprogram. 2014; 16: 428–438.
5. Diederichs S., Tuan RS. Functional comparison of human-induced pluripotent stem cell-derived mesenchymal cells and bone marrow-derived mesenchymal stromal cells from the same donor. Stem Cells Dev. 2014; 23:1594-1610.
6. Fan YX, Gu CH, Zhang YL, Zhong BS, Wang LZ, Zhou ZR, Wang ZY, Jia RX, Wang F. Oct4 and Sox2 overexpression improves the proliferation and differentiation of bone mesenchymal stem cells in Xiaomeishan porcine. Genet Mol Res. 2013; 12: 6067-6079
7. Although the manuscript is on MSCs from male rats, data presentation, especially on MSCs characterization, is very similar to: Dzobo K, Turnley T, Wishart A, et al. (2016a). Fibroblast-Derived Extracellular Matrix Induces Chondrogenic Differentiation in Human Adipose-Derived Mesenchymal Stromal/Stem Cells in Vitro. Int J Mol Sci 17. Authors would benefit from using such published manuscript for guidance when presenting their results. And should reference such papers.
We thank the reviewer for the suggestion.
This article is helpful for understanding of the differentiation potential of MSCs. We have added it to the manuscript according to the reviewer’s suggestion. The revision is highlighted in red in the manuscript.
8. Figure 3B, D. Mettl-3-sh3 did not affect Mettl3 protein levels very much yet affected osteogenic differentiation in the same way as Mettl3-sh1. Please authors must comment on this in the manuscript.
To explore the function of Mettl3 in the osteogenic differentiation process of BMSCs, the cells were transfected with specific shRNAs (#sh1, #sh2, and #sh3). The results showed that Mettl3-sh1 resulted in the most efficient silencing and thus utilized in the following experiments. Mettl3-sh3 in figure 3D was a clerical error. We apologize for this negligence and have amended it in the manuscript. Thank you for your careful work.
Round 2
Reviewer 1 Report
The Manuscript can be published in the present form